# Effectiveness and Safety of Tecneplase vs. Alteplase in the Acute Treatment of Ischemic Stroke

**DOI:** 10.3390/jpm12091525

**Published:** 2022-09-17

**Authors:** Ángel Estella, Miriam Pérez Ruiz, Juan José Serrano

**Affiliations:** 1Intensive Care Unit, University Hospital of Jerez, Medicine Department, University of Cádiz, 11407 Jerez, Spain; 2Intensive Care Unit, University Hospital of Puerto Real, 11510 Puerto Real, Spain; 3Cardiology Unit, University Hospital of Jerez, 11407 Jerez, Spain

**Keywords:** stroke, thrombolytic therapy, tenecteplase, alteplase

## Abstract

Not all hospitals have interventional radiology services. This fact implies that in centers where this resource is not available, the treatment of stroke in the acute phase must be adapted and individualized. The aim of the study is to determine and compare the combined effect of thrombolysis and thrombectomy effectiveness and safety of tenecteplase versus alteplase in the acute treatment of ischemic stroke in patients who are candidates for endovascular therapy according to clinical practice guidelines. This paper details a retrospective multicenter cohort study of patients with ischemic stroke admitted in three hospitals between 2018 and 2020. The main outcome variables were the degree of recanalization and the functional outcome at 3 months; safety variables were mortality and the occurrence of intracranial hemorrhage (ICH). In total, 100 patients were included, 20 of which were treated with tenecteplase (TNK) and 80 with alteplase (rtPA). Of those treated with TNK, 75% obtained a successful recanalization compared to 83.8% in those treated with rtPA (OR 0.58; 95% CI 0.18–1.88; *p* = 0.56). No differences were found in obtaining an excellent functional result at 3 months (35% TNK vs. 58.8% rtPA; *p* = 0.38). Tenecteplase showed worse neurological results after 24 h (unfavorable result of 70% with TNK vs. 45% with rtPA; OR = 5.4; 95% CI 1.57–18.6). No significant differences were identified in mortality; 17.5% with rtPA and 20% with TNK (*p* = 0.79), nor in the appearance of intracranial hemorrhage ICH (15.2% with rtPA vs. 30% with TNK (*p* = 0.12). In our series, there were not significant differences shown regarding effectiveness and safety between tenecteplase and alteplase.

## 1. Introduction

Stroke is a disease of great importance due to its high mortality rate and its high socio-sanitary burden [1]; two thirds of the patients who survive strokes present some kind of long-time disability. Likewise, the effectiveness of its resolution is time-dependent and its delay is associated with a worse prognosis, hence the importance of early detection and treatment. Currently, the most commonly used fibrinolytic drugs are alteplase (rtPA), the only one approved by the Food and Drug Administration (FDA) for the acute management of ischemic stroke [2,3,4], and tenecteplase (TNK), the drug of choice in acute coronary syndrome [5]. However, various pharmacological characteristics make TNK a drug to be considered as an alternative to rtPA in stroke patients. In vitro studies have shown that it is more powerful in dissolving thrombi [6], it has more affinity with fibrin, which reduces the appearance of side effects, and it is more resistant to the plasminogen activator inhibitors, which gives a longer half-life allowing for its administration in a single bolus [2,4,7,8]. For all these reasons, TNK theoretically has the potential to be pharmacologically superior in efficacy and safety compared to rtPA, and numerous studies have been carried out to transfer this superiority into clinical practice with promising results [4,7,8,9,10,11,12,13,14,15]. As a conclusion to the different clinical trials carried out, the American Heart Association/American Stroke (AHA/ASA) guidelines have recently added new recommendations for the use of TNK, indicating it as an option over rtPA in patients without contraindications for intravenous fibrinolysis whom are also candidates for mechanical thrombectomy [16]. Not all hospitals have interventional radiology service. This fact implies that in centers where this resource is not available, the treatment must be personalized. In a time-dependent disease in which important asymmetries have been documented in the treatments offered [17], it is very important not to deprive patients of the best treatment options adapted individually to the conditions of the health center where they are being treated. The aim of this study is to determine and compare the effectiveness and safety of tenecteplase versus alteplase in the acute treatment of ischemic stroke in patients who are candidates for endovascular therapy.

## 2. Materials and Methods

We conducted a multicenter, retrospective and observational cohort study in which included patients were treated in the Emergency Departments (ED) of 3 s-level hospitals with diagnosis of ischemic stroke due to large-vessel occlusion and indication for fibrinolysis plus thrombectomy, so they required a transfer to a fourth reference hospital. The study period was between January 2018 and January 2021, both inclusive. Two groups were formed according to the thrombolytic used in the acute phase of stroke: group 1, patients treated with alteplase 0.9 mg/Kg, up to a maximum of 90 mg, initially administered 10% of the dose as an intravenous bolus, and then the rest of the dose as an intravenous infusion over 60 min; and group 2, treated with tenecteplase administered by a single intravenous bolus of 0.25 mg/kg, maximum 25 mg. The inclusion criteria were: patients ≥ 18 years old with a diagnosis of ischemic stroke and who had received systemic thrombolysis and subsequent endovascular thrombectomy due to large vessel occlusion: middle cerebral artery (MCA—sphenoidal segment or M1 and insular segment or M2), carotid artery (ICA) or basilar artery seen on CT or MRI. Were excluded patients with a pre-stroke disability (mRS ≥ 3), terminally ill patients with expected survival of less than 3 months, those with significant head injury, those having had a prior stroke in the previous 3 months, and pregnant women. We classified dependent variables into effectiveness and safety variables. Effectiveness variables were: the degree of reperfusion of the ischemic territory after fibrinolysis and thrombectomy (mTICI), the value of the modified Rankin scale at 90 days after thrombolytic application (mRS90), the differential value of NIHSS scale at 24 h (ΔNIHSS24-0) of thrombolytic application compared to the premorbid value (NIHSS0) and number of days of hospital stay. The first three were dichotomized so that successful recanalization was considered as TICI 2b-3, excellent result as mRS90 0–1 and significant neurological improvement as NIHSS24 = 0 or reduction of 4 or more points compared to the baseline NIHSS prior to thrombolytic treatment. Safety variables were death from any cause within 90 days after the thrombolytic treatment and the presence of intracranial hemorrhage (ICH) in the first 48 h. The latter defined according to the European Cooperative Acute Stroke Study (ECASS) as: (i) blood at any site in the brain on the CT scan, clinical deterioration or adverse events indicating clinical worsening (drowsiness, increase in hemiparesis) or causing an increase in the NIHSS score of 4 or more points (ECASS II criteria); (ii) any apparently extravascular blood in the brain or within the cranium associated with clinical deterioration (defined by an increase in the NIHSS score of 4 or more points) or death, and that is identified as the predominant cause of the neurological deterioration (ECASS III criteria) [18]. Independent variables included were: age, gender, event-treatment time, occlusion location, premorbid mRS (mRS0), NIHSS0, antiplatelet therapy, and number of cardiovascular risk factors (CVRF) such as atrial fibrillation (AF), arterial hypertension (AHT), diabetes, dyslipidemia and smoking. 

### Statistical Analysis

SPSS v25.0 was applied for statistical analysis; based on previous literature, our goal was to detect 35% more of successful recanalization (mTICI 2b-3) in subjects treated with TNK versus those treated with rtPA (r1 = 0.50; r2 = 0.15), with a confidence level of 95%. For a sample size of 100 patients, a power of 83% was obtained. Qualitative variables were represented by absolute value and percentage. The analysis of the quantitative variables has been carried out both in a general and stratified ways by the treatment group. We performed normality tests on these variables (Kolmogorov-Smirnof test for rtPA group and Shapiro-Wilk for TNK group), demonstrating non-normal distribution, so the median was used as a measure of central tendency and the interquantile range (IQR) as a measure of dispersion. The analysis of the relationship between qualitative variables was carried out using Pearson’s Chi square, using Yates’ correction when founding at least one expected value <5, and the Odds Ratio (OR) to quantify this association. Mann-Whitney U test was used to analyze the association between quantitative and qualitative variables. For the variables mRS90 and ΔNIHSS24-0, binary logistic regressions were used to control the confounding factors.

## 3. Results

We analyzed 100 patients, (44 females), treated in the ED meeting the admission criteria; 20 of which were treated with tenecteplase and 80 with alteplase. These patients had a median age of 73 (IQR: 66.25–80.5) and a median of 2 cardiovascular risk factors (CVRF) (IQR: 1–3), the most frequent being hypertension (68%), followed by dyslipidemia (50%). Most did not show any type of disability before the onset of the event (93% mRS0 ≤ 1). Regarding the location of the occlusion, we found that the involvement of the middle cerebral artery in its M1 segment accounts for more than half of the total (55%), followed by tandem occlusion (24%); 75% were treated with fibrinolytic therapy within 3 h or less from the last time the patient was seen asymptomatic, and the median NIHSS0 was 16 points (IQR: 10–21). In the stratified analysis of the independent variables according to the treatment group, we found no statistically significant differences, with the exception of gender, with 20% of women in the TNK group (*p* = 0.016), and NIHSS0, with a median of 6 points higher in those treated in that group (*p* = 0.021). The clinical and epidemiological characteristics of the patients and the differences between groups are shown in Table 1.

### 3.1. Analysis of Effectiveness

Overall, 82% of patients achieved a successful recanalization of the vessel (mTICI 2b-3) after thrombectomy. However, only half (50%) presented a significant neurological improvement at 24 ± 6 h after the application of the thrombolytic. Patients were hospitalized for a median of 8 days (IQR: 5–15). After 90 ± 7 days of the application of the thrombolytic, 54 patients (54%) did not present any type of disability, thus obtaining an excellent result (mRS90 ≤ 1). When analyzing the differences in the degree of reperfusion of the ischemic territory, no statistically significant differences (*p* = 0.56) were detected between both treatment groups (rtPA 83.8% vs. TNK 75%). In the study of hospital stay, no significant differences were found (*p* = 0.21) in terms of days of admission (8 days in the rtPA group—IQR: 5–13 vs. 11 days in the TNK group—RIC: 6–20). Regarding the Rankin scale at 90 ± 7 days after thrombolytic application, although the percentage of excellent results is lower in patients treated with TNK (35% vs. 58.8%), these differences were not statistically significant (*p* = 0.38) after performing the binary logistic regression model. In our series, the most influential factors in predicting an unfavorable outcome (mRS90 2–6) were diabetes and ICH. Diabetic patients and those with ICH are about 4 times more likely to have an unfavorable outcome (Table 2). Finally, those treated with TNK had worse neurological outcomes 24 h after fibrinolysis (significant neurological improvement: TNK 30% vs. rtPA 55% (*p* = 0.008). In this way, those treated with TNK are approximately 5 times more likely to have an unfavorable result in the variable ΔNIHSS24-0 than those treated with rtPA (OR: 5.4 CI95% 1.57–18.6), regardless of the remaining confounding factors (Table 3).

### 3.2. Security Analysis

Subject’s mortality in the first 90 days was 18% (18) and the frequency of intracranial hemorrhage in the first 48 h after the application of the fibrinolytic was 18.2% (18). We did not observe statistically significant differences between the two treatment groups, neither in terms of mortality (17.5% with rtPA and 20% with TNK (*p* = 0.79) nor in the appearance of ICH (15.2% with rtPA vs. 30% with TNK (*p* = 0.12). When analyzing the population that presented hemorrhagic complication, we observed that diabetes predominated and a higher score of the NIHSS0 at admission. 

## 4. Discussion

Based on the results obtained in our work, we can conclude that there are no major differences in terms of the effectiveness nor safety of both drugs. Therefore, TNK is a good therapeutic option in centers that do not have interventional radiology services and have to transfer patients for endovascular treatment. The population studied is similar in terms of age to that published in the literature, but we have seen that our patients presented a higher proportion of comorbidities and a worse premorbid neurological status [9,10,11]. We observed that patients treated with tenecteplase had practically the same chances of achieving successful recanalization of the occluded vascular territory as those treated with alteplase. These results contrast with those published in the EXTEND-IA TNK trial [11] and by Kheiri et al. [15] in a meta-analysis of five clinical trials. In both studies, the proportion of patients in whom reperfusion was achieved is substantially lower than those found in our results, although this may be because the primary endpoint in these studies is recanalization in the first angiographic series, before the thrombectomy, and not after. Despite this, they reported that reperfusion was better in those treated with tenecteplase. In another previous meta-analysis, a higher proportion of complete recanalization was also found in these patients, with reperfusion data closer to our study [19]. The main and most recent clinical trials found no differences between both drugs in terms of evolution of the neurological status 24 h after fibrinolytic administration [10,11,12], although other meta-analyses showed that this evolution is more favorable with TNK [14,15,18]. This situation of unequal results between the main literatures could be due to the aged clinical trials included in the meta-analyses, in which a weak statistical association was demonstrated and could be due to the relatively small samples size. Therefore, although they are encouraging results, they must be treated with caution. In our case, we obtained even more disparate results. Patients treated with TNK had a worse neurological outcome at 24 h, regardless of baseline status and cardiovascular risk factors, with a 5-fold greater probability of not having frank neurological improvement. As opposed, we have not found statistically significant differences in the degree of medium-term disability. Despite this, patients treated with TNK have presented a worse functional status in the medium term, although these results may be due to other comorbidities (diabetes, age difference, previous neurological status or the appearance of intracranial hemorrhages at 48 h), rather than by the thrombolytic treatment. In other literature studies with similar characteristics carried out in different regions, no statistically significant differences have been found either. However, and unlike our results, the rates of functional independence at 90 days were similar between the treatment groups [9,10,11,12,14,15,19]. Finally, in terms of effectiveness, patients treated with TNK were hospitalized approximately 3 days longer than those treated with rtPA, although without statistical significance. In addition, there are currently no studies comparing the number of days of hospital stay based on the fibrinolytic administered, so we cannot draw great conclusions in this regard. The evaluation of the safety of TNK has been addressed by numerous studies and has been a common point among them. Consequently, they all show similar rates of mortality and bleeding events in both groups, with no significant differences [9,10,11,13,14,15,19]. These results are quite consistent with ours, because we have not found significant differences in the proportions of intracranial hemorrhages or mortality. However, it should be noted that in our series, we have detected higher mortality and incidence of bleeding than in the aforementioned studies. In the case of ICH, the rates in previous literature are around 5–10%, while in our series, we found notably higher rates, reaching 15% in those treated with rtPA and up to 30% in those treated with TNK. 

It is likely, although it has not been the subject of this study, that the higher incidence of bleeding complications and mortality described in our series is motivated by multiple reasons: more restrictive and controlled inclusion criteria of clinical trials, our patients presented more comorbidities and a worse baseline neurological status, heterogeneity in the definition of ICH, small sample size in our series or the need to transfer to another hospital due to the absence of an interventional radiology service in the origin center. The fact that we have found more hemorrhages in the TNK group, could also be due to the applied dose, which we are unaware of. Since there is no standardized dose of TNK, patients could have received very different doses from each other, and higher doses seem to imply an increased bleeding risk. In regard to the limitations of the study, it should be noted that its retrospective nature prevents having control over the quality of information, since we used data collected for other purposes. It is also not possible to establish a causality or risk relationship; therefore, the results obtained can hardly be considered definitive when addressing causal relationships. In addition, the study sample size is small, especially for the cohort treated with TNK, a fact that favors the influence of random chance on the results. Finally, the clinical differences are probably influenced by the outcome of thrombectomy, given its impact on proximal arteries recanalization. As a conclusion, we have found that there are no large differences in terms of effectiveness and safety between tenecteplase and alteplase, however, the ability to administer TNK as a bolus, instead of as an hour intravenous infusion, may offer benefits over rtPA. Therefore, the use of TNK could reduce administration times, and thus transfer to a reference hospital could be shortened for patients who require thrombolysis followed by endovascular treatment. Additionally, the transfer would be eased by not requiring an exclusive venous access (VA), thus avoiding the risk of extravasation or loss of the VA. It would also avoid the possibility of withdrawal of the fibrinolytic treatment in case of oscillations in blood pressure during the infusion. Therefore, the demonstration of non-inferiority, even in the absence of superiority, place TNK as an attractive alternative. Our study opens the door to new lines of research which would imply a continuity in the analysis of these indicators and new ones, such as the most appropriate dose, time saved in the administration of the fibrinolytic agent, or the facilities that the administration of TNK involves. The off-label use of TNK that we present in real clinical practice has just been endorsed with the recently published results of the AcT study [20] that provides solid evidence that tenecteplase is equally safe and effective as alteplase. These findings give light to and offer a new time for action and modification of the current intravenous thrombolysis guidelines and protocols [21].

## Figures and Tables

**Table 1 jpm-12-01525-t001:** Clinical-epidemiological characteristics of patients with ischemic stroke who are candidates for thrombolysis and thrombectomy.

	Total *n* = 100 *n* (%)	rtPA *n* = 80 *n* (%)	TNK *n* = 20 *n* (%)	*p*
**Gender (female)**	44 (44)	40 (50)	4 (20)	** *0.016* **
**Age (years) [median (IQR)]**	73 (66.25–80.5)	74.5 (66–81.75)	73 (69–78.5)	0.766
**Number of CVRF**	2 (1–3)	2 (1–3)	2 (1–3)	0.757
**Arterial hypertension**	68 (68)	55 (68.8)	13 (65)	0.75
**Dyslipidemia**	50 (50)	42 (52,5)	8 (40)	0.317
**Diabetes**	28 (28)	22 (27.5)	6 (30)	0.824
**Smoking**	18 (18)	14 (17.5)	4 (20)	1
**Atrial fibrillation**	10 (10)	9 (11.3)	1 (5)	0.677
**Antiplatelet therapy**	31 (31)	24 (30)	7 (35)	0.665
**Premorbid mRS ≤ 1**	93 (93)	75 (93.8)	18 (90)	0.14
**NIHSS0 (points) [median (IQR)]**	16 (10–21)	13.5 (9–20)	19.5 (14.75–22)	** *0.021* **
**Event-treatment time ≤ 3 h**	75 (75)	59 (73.8)	16 (80)	0.564
**Occlusion location**				0.45
**MCA M1**	55 (55)	46 (57.5)	9 (45)	
**Tandem**	24 (24)	19 (23.7)	5 (25)	
**Carotid artery (ICA)**	11 (11)	8 (10)	3 (15)	
**ACM M2**	5 (5)	4 (5)	1 (5)	
**Basilar artery**	5 (5)	3 (3.7)	2 (10)	

CVRF: cardiovascular risk factor; mRS: modified Rankin Scale; MCA: middle cerebral artery.

**Table 2 jpm-12-01525-t002:** Logistic regression model mRS90.

Variable	B	Standard Error	Wald	gl	Sig	Exp(B)	IC_95%_
**Diabetes**	1.428	0.556	6.601	1	0.010	4.171	1.403–12.4
**ICH**	1.469	0.680	4.664	1	0.031	4.344	1.145–16.48
**Age**	0.056	0.026	4.766	1	0.029	1.058	1.01–1.11
**NIHSS_0_**	0.063	0.039	2.580	1	0.108	1.065	0.986–1.15
**Days of hospital stay**	0.058	0.028	4.172	1	0.041	1.060	1.002–1.12
**Constant**	−6.503	2.079	9.786	1	0.002	0.001	

Model indicators. Log Likelihood-2: 102.248. Cox and Snell’s R2: 0.27. Nagelkerke’s R2: 0.36.

**Table 3 jpm-12-01525-t003:** Logistic regression model ΔNIHSS_24–0_.

Variable	B	Standard Error	Wald	gl	Sig	Exp(B)	IC_95%_
**Treatment group**	1.686	0.631	7.135	1	0.008	5.398	1.57–18.6
**Arterial hypertension**	0.909	0.516	3.110	1	0.078	2.482	0.9–6.8
**Diabetes**	1.273	0.530	5.765		0.016	3.572	1.264–10.1
**Smoke**	1.318	0.648	4.131	1	0.042	3.736	1.05–13.32
**NIHSS_0_**	0.097	0.038	6.692	1	0.010	1.102	1.024–1.19
**Constant**	−5.103	1.254	16.573	1	0.000	0.006	

Model indicators. Log Likelihood-2: 115.678. Cox and Snell’s R2: 0.205. Nagelkerke’s R2: 0.273.

## Data Availability

Data supporting reported results are available if is required. The duly anonymized data are guarded by the research team.

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
