# Peer review of "Effectiveness and Safety of Tecneplase vs. Alteplase in the Acute Treatment of Ischemic Stroke"

_jpm, 2022, doi:10.3390/jpm12091525_

Round 1
Reviewer 1 Report
Dear authors
if I got it right the main message of your work is that TNK should be used in those Centers that don't have an Interventional Neuroradiology service (ie the spoke centers?) because it has some advantages over rTPA for those patients that should be transferred to the Hubs. If that's the case you should underline this idea and make it the core of your paper (whereas it's reported just at the end of your "discussion" section. Using the efficacy index of the TICI score after fibrinloyisis AND thrombectomy is interesting and I appreciated it as it's the final goal of our treatment strategy as a whole.
I would create a separate "Statistic Analysis" paragraph instead of including it in the "Materials and methods" one. In the same way I think that in the "Results" section, for the sake of clarity, you should create separate paragraphs for the "Analysis of effectiveness" and "Security Analysis"
Author Response
Dear Reviewer. First of all thank you for accepting the revision of the manuscript and for your comments about it. I am convinced that your suggestions will improve the content of the manuscript. Below I respond point by point to your recommendations:
- In accordance with your recommendation we have modified the text with the idea of emphasizing the advantages of the use of tecneplase in the treatment of stroke considering the use of the efficacy index of the TICI score after fibrinloysis and thrombectomy.
Paragraph like “ Therefore, TNK is a good therapeutic option in centers that do not have interventional radiology services and have to transfer patients for endovascular treatment.” has been added in the discussion section.
- Following your instructions we have modified the methods section contemplating separately the statistical analysis.
- According with your suggestions in the Result section we have created separate paragraphs for “Analysis of effectiveness” and “security”. No doubt thanks to his suggestion the manuscript is expressed more clearly.
Additionally.
Interestingly, during the preparation of the manuscript and the review carried out by you, the clinical trial that gives the greatest support to our findings has been published in LANCET. The editorial line of the journal report according with your comments on the future of this treatment.
To documment clinical experiences may be more pertinent to know real clinical practice. I add to the manuscript the references of the clinical trial and the commentary published in Lancet (references 20 and 21)
Reviewer 2 Report
The study presented by Estella et al. reports on retrospective comparison of effectiveness and safety of thrombolysis with Tenecteplase versus Alteplase before mechanic thrombectomy in a drip-and-ship scenario in 3 contributing hospitals. In a population of 20 Tenecteplase- versus 80 Alteplase-treated, mostly highly affected patients with large vessel occlusion, the authors could not observe significant differences in thrombectomy result or medium-term clinical outcome.
Being a central curative therapeutic option for acute stroke patients, diversifying the pharmacological repertory especially of non-thrombectomy centers is crucial and therefore respective clinical experience on thrombolytics should be reported. Nevertheless, the presented manuscript has major flaws which should be improved before consideration for publication.
11) The abstract in its current form is misleading and should be completed (proportion of the treatment groups, statistical results not only for recanalization and clinical outcome but also for ICH). It should be made more clear that the results shown focus on the combined effect of thrombolysis and thrombectomy instead of directly comparing the reperfusion potential of the two. The repeated claim for personalized medicine is not supported by the study or discussed any further and should therefore – although being submitted to Journal of Personalized Medicine – be reconsidered.
22) The methods and results section should be completed by information about Alteplase and especially Tenecteplase dosage and main procedural parameters within the different treatment groups (door-to-needle, onset-to-groin time). To give relative information about the rather imbalanced overall patient population does not suffice as e.g. 75% of early-treated patients could well exclude the complete Tenecteplase group.
33) The statistical reports include some very surprising results which might be explicable by high variation within the small Tenecteplase group and should be interpreted carefully. Although logistic regression apparently did not show any correlation between initial clinical affection and final clinical outcome, a difference of 6 points in NIHSS score is subtantial and from my point of view the most plausible explanation for the differences in outcome observed. Although an ICH rate of 30% versus 15% seems not to reach significance in this dataset, it should not be treated as irrelevant. As ICH is the most feared complication of thrombolysis, observation of hemorrhage in every third patient is alarming and should cause profound re-evaluation of this treatment option, especially as the NOR-TEST 2 trial was stopped early due to observation of higher ICH rates in patients with moderate to high NIHSS scores [Kvistad et al. Lancet Neurology, May 2022]. The authors should rather use the obtained dataset to characterize the patient proportion suffering from ICH than to understate its relevance.
44) Especially the first half of the manuscript needs some editing (removing typos, unifying font size).
The authors provide a comprehensive discussion of the study’s limitations but do not seem to draw necessary consequences for the interpretation of their data. I suggest to re-write the manuscript with a stronger focus on which insights this study can actually add to the known literature (focus on elderly, comorbid patients, identification of risk factors for ICH after Tenecteplase treatment to be able to personalize choice of thrombolytic in the future).
Author Response
Dear Reviewer. First of all thank you for accepting the revision of the manuscript and for your comments about it. I am convinced that your suggestions will improve the content of the manuscript.
Our intention is to document real clinical practice based on the recommendations derived from clinical trials aware of the difficulties that the only previous therapeutic option involves continuous perfusion for centers that do not have interventional neuroradiology and have to transfer patients.
Having other options helps the clinician in making decisions.
Below I respond point by point to your recommendations:
- Abstract
According to your recommendation the abstract has been modified and completed providing new data that you suggested.Proportion of tretament groups has been added. Suggested results by reviewer have been added and completed.
Repeated claim for personalized medicine has been corrected in the text. We believe that with his accurate suggestions the abstract is clearer and more concise.
We have added information about alteplase and tenecteplase dosage according with use of medication in clinical trial: EXTEND-IA TNK: Campbell, et al. Tenecteplase versus Alteplase before Thrombectomy for Ischemic Stroke. N Engl J Med 2018;378:1573-82.
- In the analysis of the results we discuss the findings and agree with your comments. We have not underestimated the relevance of intracranial hemorrhage, results are explained by the variety of clinical manifestation and significance they have had. Indeed, as you refer might be explainable by high variation within the small tenecteplase group, this is what we explain in the limitations section.
- Following your instructions first half of the manuscript has been corrected. Some changes have been made to the discussion trying to satisfy the questions about the orientation of the text you suggest.
Interestingly, during the preparation of the manuscript and the review carried out by you, the clinical trial that gives the greatest support to our findings has been published in LANCET. The editorial line of the journal report according with your comments on the future of this treatment.
To documment clinical experiences may be more pertinent to know real clinical practice. I add to the manuscript the references of the clinical trial and the commentary published in Lancet (references 20 and 21)
Round 2
Reviewer 1 Report
Dear authors
I have appreciated your changes. I think that the abstract presents some editing errors.
Author Response
Thank you for your assessment, your contribution to the review with your comments has been very useful to the authors to improve the manuscript.
We have corrected editing errors of the abstract according with your suggestion.
Reviewer 2 Report
Dear authors,
My impression is that the manuscript has substantially improved. Nevertheless, I would ask the you to characterize briefly the subpopulation of patients who suffered from ICH in this study, as I feel that this would truely enhance the scientific relevance of the paper, given the fact that real world data should actually help clinicians making a valid choice.
Author Response
Thank you for your assessment, your contribution to the review with your comments has been very useful to the authors to improve the manuscript.
Following your instructions we have added the requested data that characterize the population that presented cerebral hemorrhage as a complication. Considering the length of the text we have added data in results section of the manuscript.
We thank you very much for helping to improve the quality of the manuscript